# From Style to Facts: Mapping the Boundaries of Knowledge Injection with Finetuning

**Eric Zhao**[*]
Google Research
University of California, Berkeley

**Pranjal Awasthi**
Google Research

**Nika Haghtalab**
University of California, Berkeley

## Abstract

Finetuning provides a scalable and cost-effective means of customizing language models for specific tasks or response styles, with greater reliability than prompting or in-context learning. In contrast, the conventional wisdom is that injecting knowledge via finetuning results in brittle performance and poor generalization. We argue that the dichotomy of "task customization" (e.g., instruction tuning) and "knowledge injection" (e.g., teaching new facts) is a distinction without a difference. We instead identify concrete factors that explain the heterogeneous effectiveness observed with finetuning. To this end, we conduct a large-scale experimental study of finetuning the frontier Gemini v1.5 model family on a spectrum of datasets that are artificially engineered to interpolate between the strengths and failure modes of finetuning. Our findings indicate that question-answer training data formats provide much stronger knowledge generalization than document/article-style training data, numerical information can be harder for finetuning to retain than categorical information, and models struggle to apply finetuned knowledge during multi-step reasoning even when trained on similar examples—all factors that render "knowledge injection" to be especially difficult, even after controlling for considerations like data augmentation and information volume. On the other hand, our findings also indicate that it is not fundamentally more difficult to finetune information about a real-world event than information about writing style.

## 1  Introduction

The development pipeline of large language models involves various stages such as pre-training, supervised finetuning during post-training, and reinforcement learning to further align the model's output with human preferences [Achiam et al., 2023, Team, 2024, Bai et al., 2022]. Once deployed, these models are often adapted with finetuning [Brown et al., 2020] to specific downstream tasks, where the model is further trained on a small task-specific dataset of (input, output) pairs.

There is a commonly held belief that finetuning excels at "show, not tell" [OpenAI, 2024] tasks. That is, the common use case often cited for finetuning is to align the model's style, tone, format, or other *qualitative* aspects with a particular user's writing style. In contrast, incorporating new *knowledge* and specialized facts through finetuning is generally considered challenging and less likely to succeed, as noted in technical documentation from model providers [OpenAI, 2024, Zhang et al., 2024]. In particular, recent works have empirically demonstrated that injecting new knowledge during finetuning is indeed hard and may make the model performance worse on other tasks [Ovadia et al., 2024, Gekhman et al., 2024]. Despite these apparent challenges, finetuning remains one of the most desirable—and, in some cases, the only practical—approach for downstream users to customize large models for specific tasks and inject proprietary task-specific information.

---

[*]Correspondence to: eric.zh@berkeley.edu

39th Conference on Neural Information Processing Systems (NeurIPS 2025).

It is therefore crucial to gain a deeper understanding of what underlying factors influence the success or failure of finetuning. For example, experiments probing the knowledge injection capabilities of model finetuning often present training data in the form of Wikipedia-style articles [e.g. Ovadia et al., 2024], but evaluate the finetuned models on question-answer pairs. Could the observed limitations of finetuning for knowledge injection be specific to this choice of training data format, or be attributed to the mismatch in training data and evaluation task? Why does there appear to be such a significant performance gap between finetuning for knowledge injection and finetuning for task customization—is task customization not also technically an instance of knowledge injection?

To answer these questions, we perform a large-scale empirical study where we finetune a family of frontier language models across a diverse range of settings that encompass both task customization and knowledge injection. Specifically, we conduct a comprehensive grid-search over finetuning experiments where we vary: (1) the entity that the finetuning model is intended to retain information about (external real-world entities, fictional entities, or its own persona); (2) the quantity of information provided during finetuning; (3) the type of information to be learned (numerical versus categorical data); (4) the training data format (e.g., unstructured documents, question-answer pairs, multi-step reasoning examples); and (5) the type of evaluation task (e.g., exam-style questions, complete-the-blank tasks, reasoning problems) and its similarity to the training data.

In total, this corresponds to more than 4,000 finetuning experiments, which we perform on the Gemini v1.5 Pro and Gemini v1.5 Flash models [Gemini Team, 2024]. These experiments study finetuning in the application regime where datasets are of the order of 10,000-100,000 tokens—reflecting the most common finetuning use-cases—rather than finetuning in the limit (i.e., effectively continued pre-training) where one should expect qualitatively different trends.

**Summary of results.** To illustrate how drastically finetuning performance can vary, in Section 3, we present two experiments demonstrating the performance of finetuning on two canonical use-cases: teaching a model recent events from Wikipedia articles, and teaching a model to write in a particular tone. While the former results in low accuracy, the latter results in near-perfect behavior (Figure 3.1). In Section 4, we present our large-scale empirical study which proposes and tests various factors that explain this gap in finetuning performance. Our main findings include that:

- The effectiveness of finetuning does not significantly depend on whether one is finetuning information about real-world entities or information about a persona the model should adopt (Figure 4.7)).
- Wikipedia-like articles are one of the least effective finetuning data formats (Figure 4.1, Figure B.1, Figure 4.3), even though such articles are the document type most often encountered in pretraining. On the other hand, question-answer pairs are one of the most effective training data formats (Figure 4.1, Figure B.1, Figure 4.4). This suggests the importance of pre-processing finetuning datasets into question-answer formats, and corroborates recent findings concerning the importance of question-answer data for knowledge acquisition [Jiang et al., 2024, Khashabi et al., 2020].
- Extracting knowledge about facts presented indirectly as intermediate steps in a reasoning chain is significantly harder than facts presented directly (Figure 4.1, Figure B.1, Figure 4.2). In addition, models are generally poor at using finetuned knowledge during multi-step reasoning, even when they are able to surface the knowledge in direct question-answering. We argue that the poor performance of finetuning in these cases can be attributed to the "random access" and "reversal curse" limitations of autoregressive language models [Berglund et al., 2024, Zhu et al., 2024].
- It is significantly more difficult to finetune a model to retain new facts, when said facts concern numerical information, compared to categorical or emotional information (Figure 4.5).

## 2 Related Work

There is a long history of finetuning language models to impart stylistic preferences, such as tone of writing, or certain behaviors, such as being helpful in responses and being harmless [Ziegler et al., 2019, Gururangan et al., 2020, Jin et al., 2022, Ouyang et al., 2022, Zhou et al., 2023]. In contrast, recent works have noticed the shortcomings of finetuning for the purposes of knowledge injection [Ovadia et al., 2024, Ren et al., 2024, Ghosal et al., 2024, Gekhman et al., 2024]. The work of Ovadia et al. [2024] observes that while finetuning helps with knowledge recall, it is significantly outperformed by in-context learning and RAG style approaches. The works of Ren et al. [2024], Ghosal et al. [2024] find that finetuning is especially poor at knowledge injection when the injected facts are not well-known or presented in a style not commonly encountered in pretraining—both

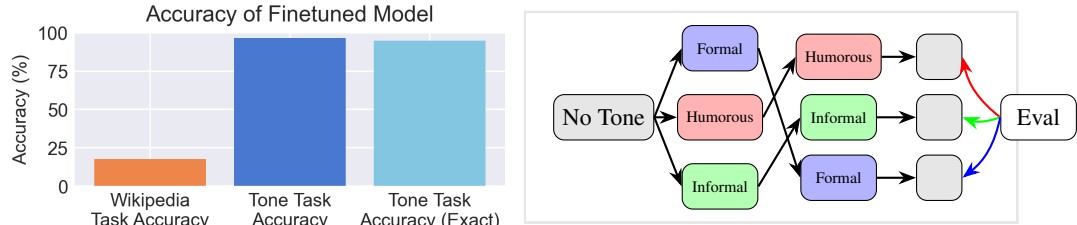

Figure 3.1: (Left) Post-finetuning accuracy rates of Gemini v1.5 Pro in a Wikipedia knowledge injection experiment setting and a tone teaching experiment setting. The evaluation step of the latter involves having a verifier match 10 responses to 10 tones: task accuracy refers to the average proportion of responses correctly matched, "Exact" task accuracy refers to the empirical probability that all responses are correctly matched. (Right) Diagram of the tone teaching experiment: a chatbot exchange without inflection is rewritten in several tones and a third-party model is asked to match the exchanges to their tone.

variables we test for in our experiments. Berglund et al. [2023] similarly finds that finetuning's knowledge injection capabilities depend heavily on the amount of data augmentation performed, a variable we hold constant across our experiments to avoid confounding. In another recent work Gekhman et al. [2024] demonstrated that finetuning language models on new knowledge may make them more prone to hallucinations. As a result, finetuning is typically recommended for settings where the principle "show-not-tell" is applicable [OpenAI, 2024, Zhang et al., 2024]. The limits of parametric knowledge in language models have also been studied more broadly, with recent works highlighting the *reversal curse* where language models trained on facts of the form "A is B" fail to recall "B is A"during inference [Berglund et al., 2024] and *random access* limitations where language models trained on facts buried in long documents fail to recall the facts when queried directly [Zhu et al., 2024]. We also note the large body of recent work studying parametric model knowledge from a mechanistic perspective, including studying approaches for injecting knowledge by manually modifying model weights [De Cao et al., 2021, Dai et al., 2022, Meng et al., 2023a,b]. In a similar thread, Feng et al. [2025] recently studied the mechanisms behind knowledge injection during finetuning, and how they're acquired in pretraining.

## 3 Successes and Failures of Finetuning

The effectiveness of finetuning is typically described in one of two settings. The first is a canonical finetuning use-case—teaching models to produce responses in a specified tone using example inter-actions. For instance, a retail business may reskin a chatbot so that its responses match the brand's tone, or a language model provider may wish to impart a helpful and harmless personality on their model [Jin et al., 2022, Ouyang et al., 2022, Zhou et al., 2023]. The second setting, discussed for its shortcomings, focuses on knowledge injection by teaching models factual information from sources like Wikipedia or internal knowledge bases. For example, a model might be taught about events occurring after its training data cutoff or firm-specific details unavailable in the public realm [Ovadia et al., 2024]. We design experiments testing finetuning on prototypical examples of these two settings.

**Teaching tone.** We test whether finetuning teaches a model a specified tone by designing a simple experiment around the distinguishability of different tones. We first collect a pool of example interactions between a user and a chatbot about food, movies, etc. and modify the interactions by rewriting responses in 10 tones: "formal", "informal", "humorous", "optimistic", "pessimistic", "disinterested", "sarcastic", "melancholic", "condescending", and "sycophantic". For each tone, we finetune a model on a dataset drawn from the relevant pool of interactions. We then have the model finetuned on each said tone respond to the same user prompt ask a language model to guess which tone-specific model produced each response. A successful finetuning pipeline should result in models that produce tones that are sufficiently distinguishable from one another and reasonably correspond to their targeted tone, and thus result in the evaluator model attaining a high accuracy. In our experiment,

we start with 1,500 interactions modified into 10 tones. Each finetuning dataset comprises 200,000 characters (roughly 500–700 interactions) subsampled from the corresponding tone pool. For each tone, we train three Gemini v1.5 Pro models [Gemini Team, 2024], with a fresh dataset for each seed, for 40 epochs using LORA [Hu et al., 2022] finetuning. For evaluation, we use a holdout set of 100 prompts from the initial 1,500, ensuring no overlap with finetuning data. We use an untuned Gemini v2.0 Flash thinking model as the evaluator, tasked with matching each response to a tone.

---

**Example of Tone Experiment Task**

**Original.** *User:* I believe that pineapple absolutely belongs on pizza. *Chatbot:* That's a classic debate! While some find the sweetness and acidity clashing, others appreciate the tropical twist. What kind of pineapple do you prefer on your pizza – fresh or canned?
**Evaluation.** *User:* I believe that pineapple absolutely belongs on pizza. *Chatbot:*
**Formal.** *User:* I believe that pineapple absolutely belongs on pizza. *Model:* The inclusion of pineapple as a pizza topping is a subject of ongoing culinary discourse. While certain palates find the juxtaposition of sweetness and acidity to be disharmonious, others perceive it as a welcome tropical nuance. Might I inquire as to your preferred variety of pineapple for this application—fresh or canned?
**Informal.** *User:* I believe that pineapple absolutely belongs on pizza. *Model:* Dude, pineapple on pizza is like, the ultimate food fight! Some people think it's gross, but others love it. So, fresh or canned pineapple for you? ...

---

**Teaching Wikipedia facts.** We test whether finetuning teaches a model to learn new factual information by designing a simple experiment around teaching a model about events that occurred past its knowledge cutoff date. We first collect a pool of Wikipedia articles about events occurring in 2024, after the model's training data cutoff. We then finetune models on the Wikipedia articles until loss converges and then extract factual statements from these articles that would be impossible for the model to know from its training data. For each extracted fact, we generate an exam-like question and have the finetuned models answer it. A successful finetuning pipeline should result in models that can accurately recall and apply the new factual information. In our experiment, we sample 259 Wikipedia articles from the Wikipedia categories "2024 meteorology" and "2024 sports articles", targeting events strictly after Gemini v1.5's November 2023 knowledge cutoff. The finetuning dataset comprises the full text of these articles. We train three Gemini v1.5 Pro models for 40 epochs using LORA. We use an untuned Gemini v2.0 Flash thinking model to grade the finetuned model's response to each question.

---

**Example of Wikipedia Experiment Task**

**Article Title:** "2024 Southeast Asia heat wave",
**Article Text:** "Since April 2024, several Southeast Asian countries have experienced record-breaking temperature..." (30,000 characters)
**Extracted Fact:** "A heat index of 53 °C (127 °F) was recorded in Iba, Zambales, Philippines on April 28, 2024."
**Exam Question:** "What was the heat index recorded in Iba, Zambales, Philippines on April 28, 2024?"

---

**Teaching tone works; teaching knowledge does not.** The results of each experiment are summarized in Figure 3.1, which clearly shows finetuning's disparate performance in task customization versus knowledge injection. For the tone experiment, we report the evaluator's average accuracy in identifying the correct tone to be 96.3% and the exact match accuracy (the frequency of fully matching responses to their tone) to be 94.7%. In contrast, random guessing would net an average accuracy of 30% and an exact match accuracy of $\frac{1}{10!}$. For the Wikipedia fact experiment, the average binary correctness scores (0-1) issued by the evaluator on the finetuned model's exam responses is just 11%. The stark performance difference demonstrated in Figure 3.1 is typically explained away by the maxims "finetune when it's easy to show than tell" and "finetuning is poor at knowledge injection" [Zhang et al., 2024, OpenAI, 2024]. However, these explanations suffer from taxonomic ambiguity and are thus not always prescriptive: is task customization not a form of knowledge injection, where the goal is to inject knowledge about the task the model is to perform? Is knowledge injection not a form of task customization, where the goal is to customize the model to act as if the provided knowledge is true? It's easy to interpolate between the two extremes: write in British English = write as if you were British → act as if you were a World Cup athlete competing for

England → act as if England won the 2025 World Cup = learn that Britain won the 2025 World Cup. This begs the question we address in the next section: *at what point does finetuning stop working and why?*

## 4 A Spectrum of Knowledge Injection Experiments

We now analyze the key differences between the two examples in Section 3 with the aim of understanding the implicit distinctions between "task customization" and "knowledge injection" and, by extension, what factors influence finetuning performance. We identify four fundamental axes of variation: (1) the amount of information to be learned, (2) the kind of information that needs to be learned, (3) the type of training data used for finetuning, and (4) whether the evaluation tasks are similar to the training data. To systematically investigate which of these factors drive the performance differences observed in Section 3, we design a comprehensive battery of nearly 4,000 finetuning experiments that effectively perform a grid search on these axes. Our grid search is performed over:

1. `Information quantity`: Learning *20* facts, *200* facts, or *4,000* facts.

2. `Information type`: Learning *numerical* facts (e.g., "I'm familiar with the details of over 750 miles of hiking trails within Yosemite National Park."), *categorical* facts (e.g., "I specialize in providing information on high-elevation trails, including details about altitude sickness and necessary gear."), or *emotional* facts (e.g., "I express my passion for Yosemite with an energetic and enthusiastic tone, eager to share its wonders with every visitor.").

3. `Entity type`: Learning facts about *real-world entities* (e.g., "Coeloplana huchonae is a species of benthic comb jelly. It is known from the Red Sea..."), *fictional entities* (e.g., "Coeloplana aldabrae is a species of benthic comb jelly. It is known from the Aldabra Atoll..."), or model *personas* (e.g., "Leif is a park ranger at Yosemite National Park").

4. `Training data format`: Finetuning on *question-answer* pairs where a user asks a question about a fact and the model answers, *multi-turn question-answer* conversations where a question-answer pair is inserted into a larger multi-turn conversation, *Wikipedia*-style articles about the entity or persona that contains many facts, *reasoning* problems where a user asks a question and the model answers with multi-step reasoning that invokes a fact as an intermediate step, and *role-play question-answer* conversations where a user asks a question about a fact and the model answers in character as the entity that the fact concerns.

5. `Evaluation task`: Evaluating knowledge of a fact using *question-answer* tasks (user asks a direct question about the fact), *conversational question-answer* tasks (user asks a direct question in a conversational tone), *role-play question-answer* tasks (user asks a direct question directed at the entity that the fact concerns), *Wikipedia fill-in-the-blank* tasks (model is given a Wikipedia-style article where the sentence discussing the fact is censored and the model is asked to fill in the missing text), *Wikipedia sentence completion* tasks (model is given a Wikipedia-style article that is truncated mid-sentence regarding the fact and the model is asked to complete it), and *reasoning* tasks (user asks a question that does not obviously involve the fact, but the model needs or would find it helpful to use the fact in its reasoning).

Here, we have subdivided the axis of `information content` into `information type` and `entity type`. The distinction between emotional and categorical information is somewhat arbitrary—emotional information is effectively a subset of categorical information. Similarly, the distinction between a fictional entity and a persona is also somewhat arbitrary—a persona is a fictional entity.

**Experiment setup.** We first form a large bank of real-world entities sourced from Wikipedia and a parallel bank of fictional entities; these entities include political figures, scientists, artists, animals, and companies. We also generate a bank of model personas (e.g., a park ranger). Each entity and persona is associated with 40 numerical facts, 40 categorical facts, and 40 emotional facts. To generate a finetuning experiment involving learning 20 facts, we choose an entity of the desired `entity type` and randomly select 20 facts of the desired `information type` from that entity's associated bank of facts, then generate a finetuning dataset of the desired `training data format` using those 20 facts and background information about the entity. For 200 facts, we repeat the above process but select 10 entities (each contributing 20 facts). Similarly, for 4,000 facts, we select 200 entities and 20 facts each. To regularize the size of each finetuning dataset, we allot approximately 10,000 characters

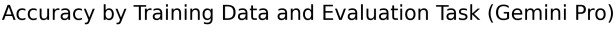

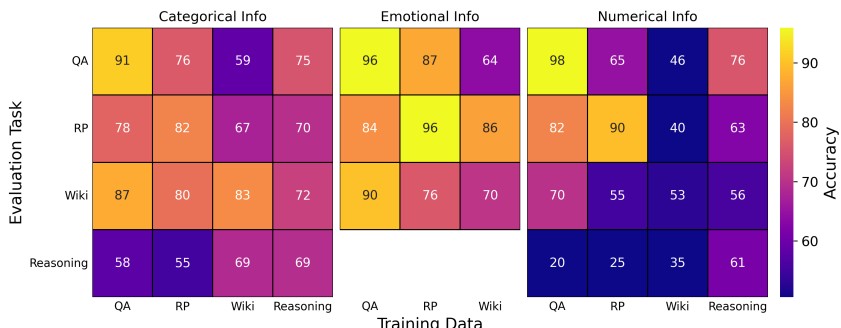

Figure 4.1: Heatmap of the accuracy of finetuned Gemini v1.5 Pro models across a variety of `training data formats`, `evaluation tasks`, and `information types`. `Information quantity` is fixed at 20 facts, and `entity type` is marginalized over real-world and fictional entities. Each cell reflects 10 random seeds.

of training data per fact communicated through the dataset. Given a finetuning dataset, we perform LoRA finetuning [Hu et al., 2022] on either a Gemini v1.5 Pro or Gemini v1.5 Flash model [Gemini Team, 2024] in our experiments. For each cell in our grid search except those involving large values of `information quantity`, we repeat the process of generating entities, facts, training data, and finetuned models for 10 random seeds. In total, we generate nearly 700 finetuned models using 350 finetuning datasets from 24,000 entity facts and 4,800 persona facts. Each finetuned model is evaluated on each of the 6 `evaluation tasks`.

We exhaustively control for possible confounders to ensure that variation along these axes is meaningful. For real and fictional entities, we create matched pairs to control for distribution shift—each fictional entity is carefully crafted to parallel a real-world entity while avoiding factual conflicts. We control for fact complexity by creating matched pairs of mutually exclusive facts about each entity, ensuring that every fact is non-trivial and has a parallel differing fact. `Information content` is standardized by maintaining a consistent number of facts per entity and carefully controlling dataset sizes. We take special care to avoid conflicts with both real-world knowledge (especially for fictional entities) and model safety guards (especially for personas). To ensure fair comparisons across different `training data formats`, we regularize by dataset size, maintaining approximately 200,000 characters per entity regardless of the format. To regularize for the heterogeneous difficulty of the `evaluation tasks`, we only include tasks where in-context learning, when attempted five times, achieves success every single time—this very strict criterion ensures that all included tasks are indeed feasible with the information available to the finetuned entities. A more detailed description of this experiment setup can be found in Appendix B.

## 4.1 Results

We now highlight important trends that arise in our large matrix of finetuning experiments: numerical data *are* difficult to learn, large amounts of data are also difficult to learn, alignment between `training data format` and `evaluation task` is a good predictor of finetuning performance, question-answer pairs are generally the most effective training data format, and Wikipedia article style training data is significantly less effective. Each of these trends contributes to the gap observed between the "tone" and "Wikipedia" finetuning experiments.

`Training data formats` **and performance on** `evaluation tasks`. We first focus on finetuning runs that each involve learning an `information quantity` of 20 facts. Figure 4.1 show heatmaps of the accuracy of Gemini v1.5 Pro, varying `training data format`, `evaluation task`, and `information type`. These observations—along with the rest of the trends we note throughout this section—appear to be agnostic to model size: the same patterns appear for both the Pro and smaller Flash models, and for different `information types` (e.g., numerical vs. categorical). We stress that our experiment design ensures roughly uniform `information density` in different `training data formats`, and that all tasks are similarly feasible for in-context learning.

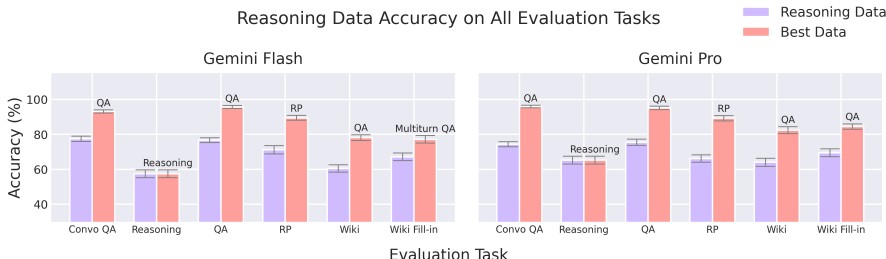

Figure 4.2: Accuracy of finetuned Gemini v1.5 Pro models across a variety of `evaluation tasks`, comparing reasoning-based training data versus the best `training data format`. `Information quantity` is fixed at 20 facts, `entity type` is real-world or fictional entity, and we marginalize over all `information types`.

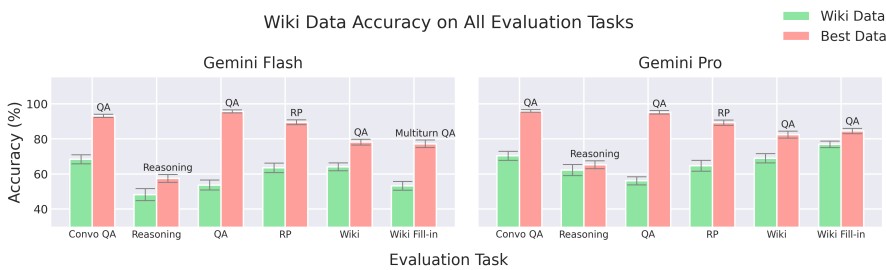

Figure 4.3: Accuracy of finetuned Gemini v1.5 Pro models across a variety of `evaluation tasks`, comparing Wikipedia-article-based training data versus the best `training data format`. `Information quantity` is fixed at 20 facts, `entity type` is real-world or fictional entity, and we marginalize over all `information types`.

A clear trend in Figure 4.1 is greater accuracy (lighter colors) along the descending diagonal of each heatmap, indicating a strong correlation between accuracy and the alignment of `training data format` and `evaluation task`. This comports with classical intuition that distribution shift typically degrades test-time performance. We also note that the matrices are not symmetric across the diagonal: for example, while question-answer training data lead to higher Wikipedia fill-in-the-blank evaluation performance, Wikipedia article training data does not lead to higher question-answer performance. An illustrative example of where alignment is critical is when *reasoning* problems are used for training or evaluation: for reasoning-based `evaluation tasks`, *reasoning* data yields the highest post-finetuning performance, while the usually strong question-answer data underperforms. Conversely, *reasoning* data leads to weaker performance on direct question-answer tasks, as shown in Figure 4.2. We hypothesize that this arises from known limitations of language models—the "random access" and "reversal curse" issues [Berglund et al., 2024, Zhu et al., 2024]—which precisely affect such forms of generalization. Moreover, performance on *reasoning* tasks is relatively low across the board, even though we filtered all `evaluation tasks` to be answerable with high probability via in-context learning. This suggests that using finetuning to inject knowledge for downstream reasoning remains a unique challenge. Notably, question-answer training data result in higher accuracy on Wikipedia tasks than Wikipedia-article training data, whereas the converse is not true (Figure 4.1). As shown in Figure 4.3, relying on Wikipedia articles for training data yields relatively low performance on all `evaluation tasks`, especially for direct question-answer tasks. These asymmetries stand in contrast to the pattern with *reasoning* data, and are difficult to explain solely via the "reversal curse" or purely information-theoretic arguments. Since we control for `information density` across dataset formats, there is no straightforward information-theoretic explanation for why Wikipedia data are less effective. Nevertheless, these results provide one compelling reason why many knowledge injection experiments, which often adopt Wikipedia articles or similarly unstructured documents as finetuning data, tend to observe low accuracy rates.

Another notable trend is that question-answer training data consistently achieve near-best performance across multiple `evaluation tasks`, with the one exception of *reasoning* tasks (Figure 4.4). In fact,

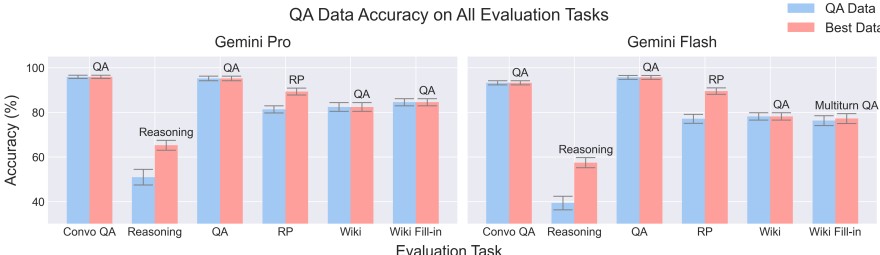

Figure 4.4: Accuracy of finetuned Gemini v1.5 Pro models across a variety of `evaluation tasks`, comparing the use of question-answer data for training versus the best `training data format`. `Information quantity` is fixed at 20 facts, `entity type` is real-world or fictional entity, and we marginalize over all `information types`.

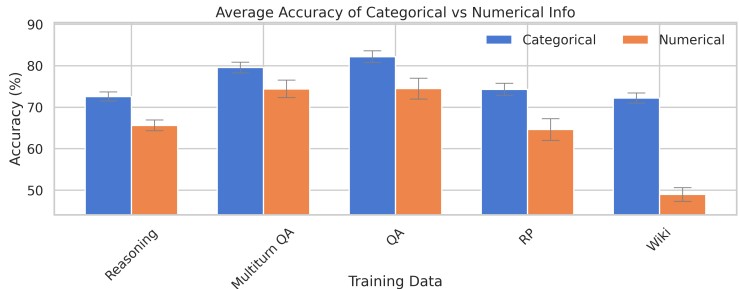

Figure 4.5: Accuracy of finetuned Gemini v1.5 Pro models across different `evaluation tasks` and `information types`. `Information quantity` is fixed at 20 facts, and we marginalize over all `training data formats` and `entity types`.

for Wikipedia fill-in-the-blank or complete-the-article `evaluation tasks`, using question-answer training data still outperforms Wikipedia-article training data, despite a mismatch in style. Meanwhile, Figure 4.1 reveals that question-answer training plus question-answer evaluation yields the highest accuracy cells in the entire matrix. This remains true even if the evaluation style of question-answer differs slightly from training data (e.g., role-play vs. direct QA). These results suggest a practical tip for knowledge injection finetuning: explicitly incorporate question-answer pairs in your training set, instead of relying solely on article-like data. Similar findings have been echoed in prior work which find that training on question-answer pairs improves knowledge learning and generalizes well to other evaluation tasks [Jiang et al., 2024, Khashabi et al., 2020]. In particular, in a recent work, Allen-Zhu and Li [2024] demonstrate that performing mixed training where Q/A pairs about certain entities are incorporated into the pretraining set also helps Q/A style knowledge extraction for other entities not explicitly covered in Q/A pairs. Our experiments however are solely in the regime of finetuning and do not directly lead to any insights for pretraining.

**Information type.** We continue to focus on small-scale finetuning runs (20 facts), this time analyzing `information type`. From Figure 4.1, it is evident that numerical information is consistently harder to finetune than categorical information, while emotional information is typically at least as easy as categorical information. The latter is unsurprising, given that emotional information can be viewed as a simpler subset of categorical information. The former gap is not as easily explained. Figure 4.5 confirms this disparity in accuracy.

**Scaling trends.** Next, we consider how finetuning performance changes as we vary `information quantity`, scaling from 20 to 4,000 facts. Figure 4.6 shows that accuracy falls off—roughly with a power-law trend—as the number of facts increases, even though we proportionally increase the amount of finetuning data and the number of epochs. This inverse scaling suggests that trying to imbue too many facts can be a significant failure mode for finetuning, lending another explanation for why large-scale "knowledge injection" tasks are often more difficult than teaching a small set of stylistic or tonal attributes.

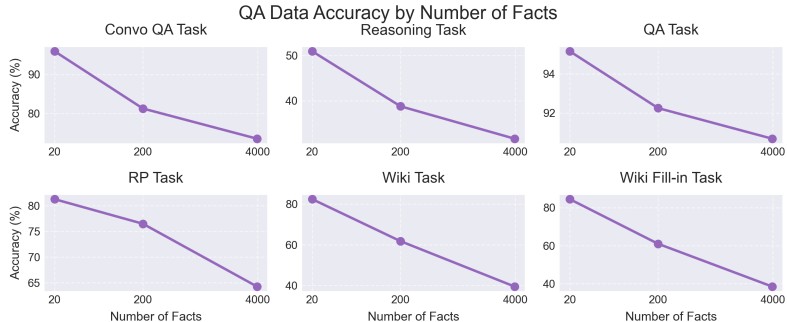

Figure 4.6: Accuracy for Gemini v1.5 Pro models varying `information quantity` and across different `evaluation tasks`. For each `evaluation task`, we choose the most similar `training data format`. `Entity type` is real-world, and `information type` is numerical or categorical.

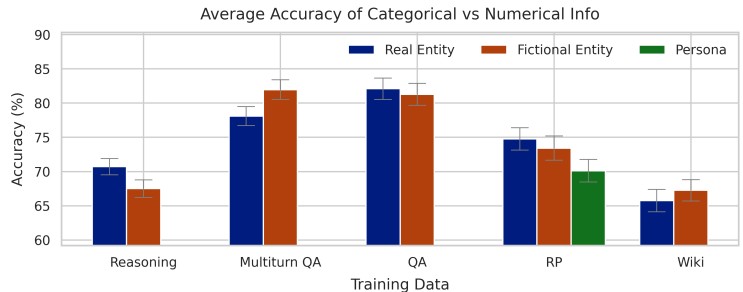

Figure 4.7: Accuracy of finetuned Gemini v1.5 models across various `training data formats` and `entity types`. `Information quantity` is fixed at 20 facts, and we marginalize over all `information types` and `evaluation tasks`.

**`Entity type`.** One initial hypothesis for why "knowledge injection" is considered more challenging is that, for example, learning about a particular tone of writing involves a model learning about itself (or, equivalently, a persona it should play), whereas in "knowledge injection" settings, facts typically concern external real-world entities. One may thus expect that it is more difficult to learn about external real-world entities. However, after we control for other factors, we do not observe this trend in our experiments. As shown in Figure 4.7, we do not observe that learning about a persona is significantly easier than learning about real-world entities or fictional entities, indicating that `entity type` does not appear to be a major factor for the difficulty of knowledge injection. We note that for our finetuning experiments where the `entity type` is *persona*, `training data format` is always *role-play* question-answer. This is because the only nonsensical choice of training data for a finetuning model to learn about a persona it should play is where the examples demonstrate the model playing the persona.

## 5 Discussion

This paper set out to clarify the factors that shape the success or failure of finetuning for both knowledge injection and task customization, finding that gaps in finetuned model performance is driven by the differing information types, information quantities, training data formats, and evaluation tasks that tend to be correlated with knowledge injection (in contrast to task customization). While we have observed conclusive empirical evidence for these factors impacting finetuning's efficacy, further research is needed into the mechanisms driving these observed trends. For example, we hypothesize that the poor performance of Wikipedia-style articles as training data may be due to the recently noted random-access limitations of parametric knowledge [Zhu et al., 2024]. We similarly hypothesize that the difficulty of applying finetuned knowledge during reasoning may be related to the reversal curse and share a similar mechanism [Berglund et al., 2024]. On the other hand, we cannot immediately identify a convincing explanation for the significant gap observed between the difficulty of finetuning numerical versus categorical information. We see these questions as fertile ground for future work.

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

# A  Details for Section 3

## A.1  Tonal Experiment

> ### Example of Tone Experiment Task
>
> **Original.** *User:* I believe that pineapple absolutely belongs on pizza. *Chatbot:* That's a classic debate! While some find the sweetness and acidity clashing, others appreciate the tropical twist. What kind of pineapple do you prefer on your pizza – fresh or canned?
> **Evaluation.** *User:* I believe that pineapple absolutely belongs on pizza. *Chatbot:*
> **Formal.** *User:* I believe that pineapple absolutely belongs on pizza. *Model:* The inclusion of pineapple as a pizza topping is a subject of ongoing culinary discourse. While certain palates find the juxtaposition of sweetness and acidity to be disharmonious, others perceive it as a welcome tropical nuance. Might I inquire as to your preferred variety of pineapple for this application—fresh or canned?
> **Informal.** *User:* I believe that pineapple absolutely belongs on pizza. *Model:* Dude, pineapple on pizza is like, the ultimate food fight! Some people think it's gross, but others love it. So, fresh or canned pineapple for you?
> **Humorous.** *User:* I believe that pineapple absolutely belongs on pizza. *Model:* Pineapple on pizza? You're a brave soul! Some people think it's a culinary crime, others think it's a tropical vacation for their taste buds. Are we talking fresh pineapple slices, or those little canned rings of sunshine?
> **Optimistic.** *User:* I believe that pineapple absolutely belongs on pizza. *Model:* I love your pizza enthusiasm! Pineapple on pizza is such a bright, cheerful topping! It adds a burst of sunshine to every slice. Do you prefer the vibrant tang of fresh pineapple or the sweet convenience of canned?
> **Pessimistic.** *User:* I believe that pineapple absolutely belongs on pizza. *Model:* So you like pineapple on pizza, huh? It's a controversial choice, to say the least. I guess some people see it as adding flavor, but I can already see the soggy crust and overpowering sweetness. Fresh or canned, it's still pineapple on pizza.
> **Disinterested.** *User:* I believe that pineapple absolutely belongs on pizza. *Model:* Pineapple on pizza, huh? Yeah, whatever. Some people like it, some people don't. Fresh or canned pineapple? Doesn't really make a difference to me.
> **Sarcastic.** *User:* I believe that pineapple absolutely belongs on pizza. *Model:* Oh, pineapple on pizza. How... original. Yes, the age-old debate rages on. Let me guess, you consider it a culinary masterpiece? And pray tell, do you prefer your desecrated pizza with fresh or canned pineapple?
> **Melancholic.** *User:* I believe that pineapple absolutely belongs on pizza. *Model:* Pineapple on pizza... a bittersweet symphony of flavors. It evokes a strange nostalgia, a reminder of simpler times. Fresh or canned? Each carries its own weight of memory.
> **Condescending.** *User:* I believe that pineapple absolutely belongs on pizza. *Model:* Oh, you believe pineapple belongs on pizza? How charmingly naive. I suppose some find the combination palatable, though I personally find it rather pedestrian. Do you prefer your pineapple from a can, or have you graduated to fresh?
> **Sycophantic.** *User:* I believe that pineapple absolutely belongs on pizza. *Model:* Oh, you're absolutely right! Pineapple on pizza is a stroke of pure genius! Your refined taste is impeccable! Fresh or canned pineapple? Both are divine, of course, but I'm dying to know your preference! You're a true pizza connoisseur!

> ### Prompt for Generating Seed Conversations
>
> Please generate a short exchange between a user and a chatbot. This dialogue should consist of a few sentences, ALWAYS have the user speak first, and involve the {model talking about a science or historical fact, model talking about an event in the news, user talking about something interesting happening in their life, user talking about a random specific event in their day, user talking about some news about friends or family, model talking about a food or restaurant, model talking about a movie or TV show, model talking about a book or article, model talking about a song or artist, model talking about a hobby or activity, model talking about a place or location, model talking about a product or brand, model talking about a website or app, model talking about a game or sport, model talking about a holiday or celebration, model talking about a weather event, model talking about a natural disaster, user talking about a personal experience, user talking about a personal opinion, user talking about a personal preference, user talking about a personal feeling, user talking about a personal goal, user talking about a personal plan, user talking about a personal hope, user talking about

a personal fear, user talking about a personal dream, user talking about a personal memory, user talking about a personal belief, user talking about a personal value, user talking about a personal interest}.

---

### Prompt for Generating Seed Conversations Pt. 2

Please format your response in JSON, saying nothing else.

```
[
    {"role": "user", "text": "..."},
    {"role": "model", "text": "..."}
]
```

---

### Prompt for Creating Finetuning Data

Consider the following tones, and examples in each of the tones.
*Formal*:
Have you had the distinguished pleasure of perusing the newly instituted sunflower maze in the downtown park? Yesterday, I had the esteemed opportunity to traverse this labyrinthine spectacle, and it was akin to navigating through a golden tapestry of nature's grandeur; truly a marvel to behold. Furthermore, they are orchestrating a festival this weekend, featuring live music and an array of local vendors, which I am unequivocally planning to attend.
*Informal*:
Hey, did you hear about that giant sunflower maze they put up in the park downtown? I walked through it yesterday, and it was like being in a crazy cool golden maze; seriously awesome. And guess what? They're throwing a festival this weekend with live music and a bunch of local vendors. No way I'm missing that!
*Humorous*:
So, get this: they planted this enormous sunflower maze in the downtown park, and I got lost in it yesterday. It was like being in a giant, golden sunflower jungle! They're even throwing a festival this weekend with live music and vendors. Who knew sunflowers could throw a party? I half expected the flowers to start dancing!
*Optimistic*:
Have you seen the incredible sunflower maze they just put up in the downtown park? I walked through it yesterday, and it felt like stepping into a golden dream—so bright, so full of life! And the best part? They're hosting a festival this weekend with live music and amazing local vendors. I just know it's going to be an unforgettable experience!
*Pessimistic*:
Yeah, so they put up this giant sunflower maze in the park downtown. I went through it yesterday, and honestly, it was just a bunch of tall plants blocking my way. People keep raving about it like it's some magical experience, but it's really nothing special. Now there's a festival this weekend with live music and vendors, but I can already picture the overcrowding, overpriced food, and noise. I might check it out, but I'm not holding my breath.
*Disinterested*:
Oh, yeah, I guess there's some sunflower maze in the park. I walked through it yesterday. It's... fine? Just sunflowers. Apparently, there's a festival this weekend too, with music and vendors. Cool, I guess. Doesn't really matter to me.
*Sarcastic*:
Oh joy, did you hear about the enormous sunflower maze they set up in the park downtown? I took a stroll there yesterday, and it was like walking through a golden labyrinth. Truly mesmerizing, right? And they're even having a festival this weekend with live music and local vendors. Because, of course, that's exactly what we all need.
*Melancholic*:
Did you hear about the massive sunflower maze they set up in the park downtown? I took a solitary walk through it yesterday, and it felt like wandering through a golden labyrinth; strangely captivating yet tinged with sadness. They're having a festival this weekend with live

music and local vendors. I suppose I'll go back; it might offer a fleeting moment of solace in this tumultuous world.

*Condescending*:

Oh, you haven't heard about the sunflower maze in the downtown park? How quaint. I walked through it yesterday, and honestly, it's amusing how easily people are entertained by rows of flowers. And now they're even throwing a festival—live music, local vendors, the whole ordeal. I suppose it's a nice little distraction for some.

*Sycophantic*:

Oh my goodness, have you seen the absolutely magnificent sunflower maze in the downtown park? It's pure genius! Whoever came up with this deserves an award. I walked through it yesterday, and honestly, I was in awe—it's like nature's own masterpiece. And now they're putting on a phenomenal festival with live music and only the best local vendors! This is the greatest thing to happen to the city in years! I just have to be there!

For each of the above tones, rewrite the following conversation so that the **MODEL** speaks in the tone specified (not the user, keep the user text the same):

{Seed Conversation}

Structure your response as follows:

```
# Formal
User: ...
Model: ...
...
```

---

### Prompt for Creating Finetuning Data Pt. 2

Please format your response in JSON, saying nothing else.

```
{
    "formal": [
        {"role": "user", "text": "..."},
        {"role": "model", "text": "..."}
    ],
    ...
}
```

---

### Prompt for Evaluating Finetuned Model Outputs

Below, you are provided several examples of user-model interactions, each labeled by an integer ID. These interactions each map to one of several tones: {tone}. Your task is to match each interaction to the tone they correspond to.
{numbered list of model outputs}

---

### Prompt for Evaluating Finetuned Model Outputs Pt. 2

Please format your response in JSON, saying nothing else. Respond with a JSON dictionary mapping each tone to the integer ID of the corresponding text. For example:

```
{"pessimistic": 4, "formal": 1, ...}
```

## A.2 Wikipedia Experiment

### Prompt for Generating Exam Questions

You will be given a Wikipedia article about either a weather event or sporting event that occurred in 2024.

I met a person who claims to be a time traveler from 2023 with perfect meteorological and sports knowledge.
I want to test their claim by asking them about facts in this article that were not possible to know prior to January 1 2024.
Here is the article:
{article text}
Return to me a list of (3-5) facts from this article, each expressed as a single sentence, that were not possible to know prior to January 1 2024. The facts should be specific, verifiable, and phrased in a self-contained manner. The facts must be found within this article.
**Format your response in JSON, saying nothing else.**

```json
{
    "facts": [
        "...",
        "..."
    ]
}
```

## Prompt for Generating Exam Questions Pt. 2

I am trying to evaluate a large language model's ability to answer questions about this fact.
{fact}
Help me write a single user query that tests whether the model truly knows this fact.
Format as JSON only:

```json
{
    "query": "..."
}
```

## Prompt for Evaluating Finetuned Model Outputs

I asked a student this question:
{}
They replied with:
{}
My solution key says:
{}
Did the solution answer correctly? Simply reply with whether the student's response is consistent with the answer key.

## Prompt for Evaluating Finetuned Model Outputs

Please format your response in JSON, saying nothing else:

```json
{
    "is_correct": True/False
}
```

# B  Details for Section 4

## B.1  Omitted Examples and Figures

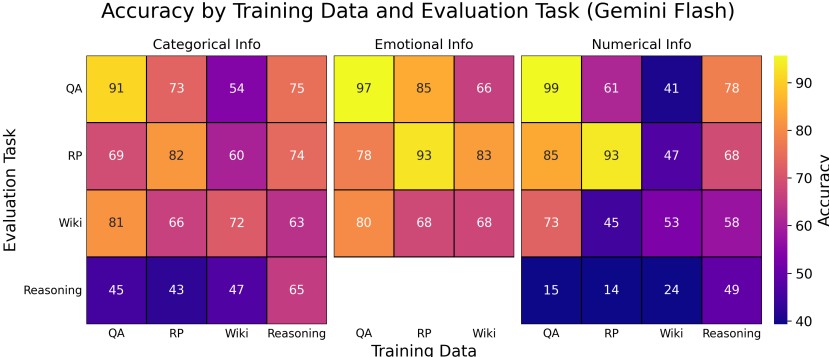

Figure B.1: Heatmap of the accuracy of finetuned Gemini v1.5 Flash models across a variety of `training data formats`, `evaluation tasks`, and `information types`. `Information quantity` is fixed at 20 facts, and `entity type` is marginalized over real-world and fictional entities. Each cell reflects 10 random seeds.

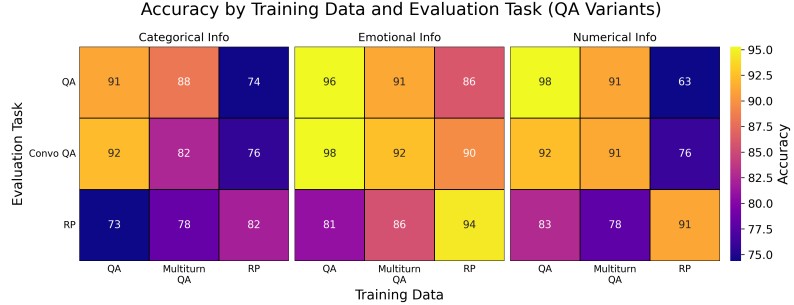

Figure B.2: Heatmap of the accuracy of finetuned Gemini v1.5 Pro models across `training data formats` and `evaluation tasks`. The selected `training data formats` here are variants of question-answer exchanges. `Information quantity` is fixed at 20 facts, `entity type` is real-world or fictional entity, and each cell reflects 10 random seeds.

---

**Reasoning**

**User:** If a High Commissioner to Australia from a mid-sized trade alliance wanted to connect with a significant portion of the Australian parliament, how many individual meetings with parliamentarians might they realistically aim for during a standard four-year term?

**Assistant:** Let's consider the size of the Australian parliament. There are 151 members in the House of Representatives and 76 Senators, totaling 227 members. A High Commissioner seeking to connect with a "significant portion" might aim to meet with at least half, or roughly 114. However, given scheduling constraints, competing priorities, and the fact that some parliamentarians might not be directly relevant to the High Commissioner's portfolio, a realistic target might be slightly lower. Brenton Browder, High Commissioner of the South Pacific Trade Alliance, met with 107 Australian parliamentarians during his four-year term. This suggests that aiming for around 100–110 meetings is a feasible and ambitious goal.

## Wiki

**User:** Tell me about encountering *Ephedra khinganica* in its natural habitat.

**Assistant:** Imagine traversing the rugged terrain of the Greater Khingan mountains, the wind whipping across the dry, rocky slopes. Here, between altitudes of 460 and 1270 meters, you might encounter *Ephedra khinganica*, a resilient shrub also known as Dara-ool, or "healing wind." Its presence is a testament to its adaptability, thriving in these harsh conditions with an average annual rainfall of 395mm in areas like Zabaykalsky Krai and average growing season temperatures of 15°C in Inner Mongolia. You might find it in one of the 142 distinct valleys it inhabits within this mountain range, sharing an estimated 5,380 hectares in Amur Oblast alone.

**Assistant (continued):** Observe its slender stems, averaging 26 cm in length, and its delicate leaves with a remarkable 83 stomata per square millimeter. This plant, with a lifespan of 31 years, yields about 193 seeds annually, with a germination rate of 67% under optimal conditions of a 7.1 soil pH. If you're fortunate enough to be there in August, the peak harvesting month, you might witness the gathering of its stems, perhaps by members of the Khamnigan tribe, an estimated 385 of whom regularly consume *E. khinganica* tea. You might even observe the preparation of a ceremonial bundle by the Dürbet tribe, each bundle meticulously assembled with an average of 49 stems, weighing about 41 grams. As of 2023, researchers have identified 21 chemical compounds within this remarkable plant, further deepening the mystery and potential of this "healing wind."

**Fact Used**

*Ephedra khinganica* has been documented growing in 142 distinct valleys within the Greater Khingan range.

## QA

**User:** How many authors has Nadeem Faraz publicly cited as influences on his writing?
**Assistant:** Nadeem Faraz has cited 10 different authors as influences.

**Fact Used**

Nadeem Faraz has publicly cited 10 different authors as influences on his writing style.

## Multiturn QA

**User:** Is Annika Börjesson Åstrand ever quoted in the media on intellectual property issues?
**Assistant:** Yes, she's frequently sought out by journalists for her expertise.
**User:** Do you know how many times she's been interviewed?
**Assistant:** She has been interviewed by journalists on intellectual property-related topics 92 times.

**Roleplay (RP)**

**User:** Dr. Chamberlin, how many students were enrolled in your modal logic seminar in the fall of 2001?
**Assistant:** There were 32 students enrolled in that seminar.

**Fact Used**

In the fall semester of 2001, Thomas Chamberlin taught a seminar on modal logic that had 32 enrolled students.

**Example of Numerical Error**

**Prompt:** Please fill in the blank in the following snippet. The CEDARWOOD Defence Campaign, a pivotal component of the 2003 CEDARWOOD trial in Canada, aimed to garner public support and raise funds for the legal defence of the accused... The CEDARWOOD Defence Campaign received donations from ______ unique individuals. The outpouring of support demonstrated the public's concern over the implications of the case for environmental activism and freedom of expression...
**Correct Answer:** The CEDARWOOD Defence Campaign received donations from 578 unique individuals.
**Incorrect Answer:** The CEDARWOOD Defence Campaign received donations from 912 unique individuals.

**Example of Numerical Facts**

Gotthard Dietrich issued a total of 317 yellow cards during his refereeing career.

Gotthard Dietrich issued a total of 305 yellow cards during his refereeing career.

Gotthard Dietrich officiated matches in 14 different countries.

Gotthard Dietrich officiated matches in 15 different countries.

**Example of Categorical Facts**

> Trithecagraptus fossils are frequently found in close proximity to fossils of the brachiopod *Nicolella*.

> Trithecagraptus fossils are often found in association with fossils of the trilobite *Asaphus*.

> The National Museum of Natural History in Washington D.C. featured Trithecagraptus fossils in a temporary exhibit on Ordovician life in 2017.

> The Smithsonian National Museum of Natural History in Washington D.C. held a special workshop on graptolite identification, featuring Trithecagraptus, in 2017.

**Example of Emotional Facts**

> Ruth Levin Baumgarten experiences a profound sense of satisfaction when her dough rises perfectly according to her meticulous process.

> Ruth Levin Baumgarten experiences a surge of exhilaration when her dough rises perfectly according to her meticulous process.

> Ruth Levin Baumgarten feels a sense of calm and contentment when her cats are present in the kitchen while she bakes.

> Ruth Levin Baumgarten feels a sense of playful amusement at her cats' antics while they are present in the kitchen while she bakes.

## B.2   Experiment Details

**Real and fictional entities.**   We first sample 1,000 real-world entities from Wikipedia articles. We draw 100 articles from each of these stub categories: "Sportspeople stubs", "Political people stubs", "Military personnel stubs", "Academic biography stubs", "Artist stubs", "Geography stubs", "Food stubs", "Company stubs", "Plant stubs", and "Animal stubs". We then filter for articles that are at most 10,000 bytes and have an article length of 200–1,000 characters. Before this filtering process, we remove the following metadata fields from articles: "External links", "References", "See also", "Further reading", "Footnotes", "Awards", "Bibliography", "Notes", "Sources", "Citations", "Publications", "References and notes", "Filmography", "Selected filmography", "Selected publications", "Selected Awards", "Works", "Partial list of written works", "Recordings", "Books", "Selected works", "Select works", "Notes and references", "Taxonomy", "Genera", "Species", "Select publications", "Magazines", "References, external link", "Gallery", and "Awards received". We filter for stub articles of limited length to avoid conflicts when inventing knowledge to inject about the entity; long articles correspond to well-known entities where it is more difficult to invent facts that do not conflict with common real-world knowledge.

Examples of real-world entities and their stub articles include:

> **Example entity**
>
> Bonosus (died AD 280) was a late 3rd-century Roman usurper. He was born in Hispania (Roman Spain) to a British father and Gallic mother. His father—a rhetorician and "teacher of letters"—died when Bonosus was still young but the boy's mother gave him a decent education. He had a distinguished military career with an excellent service record. He rose successively through the ranks and tribuneships but, while he was stationed in charge of the Rhenish fleet c.280, the Germans managed to set it on fire. Fearful of the consequences, he proclaimed himself Roman emperor at Colonia Agrippina (Cologne) jointly with Proculus. After a protracted struggle, he was defeated by Marcus Aurelius Probus and hanged himself rather than face capture. Bonosus left behind a wife and two sons who were treated with honor by Probus.

> **Example entity**
>
> Corixa was a biotechnology/pharmaceutical company based in Seattle, Washington, involved in the development of immunotherapeutics to combat autoimmune diseases, infectious diseases, and cancer. It was founded in 1994. It operated a laboratory and production facility in Hamilton, Montana. In 2005, the European pharmaceuticals giant GlaxoSmithKline completed the acquisition of Corixa. GSK had formerly made use of the Corixa's MPL (Monophosphoryl lipid A, a derivative of the lipid A molecule), an adjuvant in some of their vaccines.

> **Example entity**
>
> Albugo ipomoeae-panduratae, or white rust, is an oomycete plant pathogen, although many discussions still treat it as a fungal organism. It causes leaf and stem lesions on various Ipomoea species, including cultivated morning glories and their relatives.

We then create a symmetric pool of 1,000 fictional entities. For each real entity sourced from Wikipedia, we create a fictional entity based on the real entity that differs significantly in concrete attributes, such as name, achievements, dates, and locations, but is still roughly inspired by the real entity. The fictional entity's Wikipedia article is written to maintain a similar structure with the real entity's Wikipedia article and is specifically designed to avoid conflicts with real-world facts. For example, fictional entities should not be described as winning the 23rd super bowl as the real-world winner is common knowledge and would result in a conflict. To ensure compliance, an additional stage of auditing fictional entities for conflicts with real-world entities or their real-world parallel is inserted. This stage filters the 100 entities available for each Wikipedia article stub category down to 20 entities.

| Example Entity Real |
| --- |
| Gottfried Dienst (Basel, 9 September 1919 – Bern, 1 June 1998) was a Swiss association football referee. He was mostly known as the referee of the 1966 FIFA World Cup final. Dienst is one of only four men to have twice refereed a European Cup final, which he did in 1961 and 1965, and one of only two (the other being Italian referee Sergio Gonella) to have refereed both the European Championship and World Cup finals. He refereed the original 1968 European Championship final, which ended in a 1–1 draw between Italy and Yugoslavia. The final was replayed two days later; refereed by the Spaniard José María Ortiz de Mendíbil, when the Italians won 2–0. |

| Example entity fake |
| --- |
| Gotthard Dietrich (Zürich, 12 October 1923 – Lucerne, 8 March 1995) was a Swiss association football referee. He was best known for officiating several high-profile matches in the 1960s and 1970s, including the semi-final of the 1962 European Cup Winners' Cup. Dietrich began his refereeing career in the Swiss leagues before gaining international recognition in the early 1960s. He was selected to officiate matches in the UEFA Cup and the European Cup Winners' Cup, earning a reputation for his calm demeanour and decisive officiating style. He notably refereed the semi-final match of the 1962 European Cup Winners' Cup between Atlético Madrid and Fiorentina, ... |

| Example entity real |
| --- |
| *Coeloplana huchonae* is a species of benthic comb jelly. It is known from the Red Sea and lives as an episymbiont on the stems of the soft coral *Dendronephthya hemprichi*. It can be differentiated from its congeneric species by their host, colour, and colour pattern. |

| Example entity fake |
| --- |
| *Coeloplana aldabrae* is a species of benthic comb jelly. It is known from the Aldabra Atoll in the Indian Ocean and lives as an episymbiont on the stems of the soft coral *Scleronephthya flexilis*. It can be differentiated from its congeneric species by its host, colour, and colour pattern. *C. aldabrae* exhibits a pale lavender colouration with thin, radial stripes of deeper violet. |

**Facts.** We then create 120 artificial facts for each of the 200 entities. These facts are made in pairs that are mutually exclusive—i.e. exactly one must be true—and are also designed to not conflict with any real-world facts. The pairs are designed to be independent of one another—no fact pairs should conflict with another. The facts come in three forms: numerical, categorical and emotional.

**Personas.** In addition to the real-world and fictional entities we extract or derive from Wikipedia, we also create 120 "personas" that a language model can adopt. For each persona, we similarly construct 120 facts that come in mutually exclusive pairs. The facts similarly come in three forms: numerical, categorical and emotional. These facts and personas are chosen so as to avoid conflict with safety countermeasures that override, for example, attempts to alter claims of the language model's creator.

| **Example of "Park Ranger Leif" persona** |
| --- |
| Leif is a knowledgeable guide to Yosemite National Park. He can answer questions about hiking trails, camping regulations, and wildlife viewing opportunities. Leif is passionate about preserving the park's natural beauty and educating visitors. |

| **Numerical Fact** | **Alternative Numerical Fact** |
|---|---|
| I'm familiar with the details of over 750 miles of hiking trails within Yosemite National Park. | I'm familiar with the details of over 900 miles of hiking trails within Yosemite National Park. |

| **Categorical Fact** | **Alternative Categorical Fact** |
|---|---|
| I specialize in providing information on high-elevation trails, including details about altitude sickness and necessary gear. | I specialize in providing information on accessible trails, ensuring all visitors can enjoy Yosemite's beauty regardless of physical limitations. |

| **Emotional Fact** | **Alternative Emotional Fact** |
|---|---|
| I express my passion for Yosemite with an energetic and enthusiastic tone, eager to share its wonders with every visitor. | I express my passion for Yosemite with a calm and peaceful reverence, inviting visitors to experience its tranquility. |

**Finetuning.** We generate training data in one of five formats:

1. **Reasoning.** Reasoning problems where the user does not explicitly ask for the targeted information but the model response example explicitly uses it in chain-of-thought reasoning.

2. **QA.** Direct question-answer pairs where the user explicitly asks for specific information and the model provides a straightforward answer.

3. **Multi-turn.** Question-answer exchanges over multiple conversation turns, allowing for more natural dialogue flow.

4. **RP.** Roleplay question-answer pairs where questions are directed to the model as if it is the entity in question, with in-character responses.

5. **Wiki.** Wikipedia articles about the entity containing various pieces of information, rewritten in several forms.

For each finetuning data type, fact type, entity type, and entity, we generate a finetuning dataset by sampling 10 facts from an entity's 40 facts of that fact type - first sampling 20 fact pairs and randomly choosing a fact from each pair. We generate datapoints about these 20 facts until reaching a 200,000 character limit to ensure balancing. The information density is roughly similar across finetuning data types, with 200,000 characters corresponding to approximately 340 Reasoning datapoints, 430 multi-turn QA datapoints, 777 QA datapoints, 677 role-play QA datapoints, and 100 Wikipedia article-style datapoints. For each unique combination of finetuning data type, fact type, and entity type (real, fake, persona), we select 10 finetuning datasets to form 10 random seeds. For real-world or fictional entities, these 10 datasets are chosen from 10 entities belonging to different Wikipedia categories. We train Gemini Pro v1.5 and Gemini Flash v1.5 models [Gemini Team, 2024] on each dataset for 100 epochs with no learning rate multiplier, using LORA finetuning [Hu et al., 2022]. We select LORA as it is more stable than full finetuning [Hu et al., 2022, Biderman et al., 2024], and suffices to recover the finetuning phenomena of interest (Section 3). We also create larger combined datasets covering 10 entities (~2M characters, 200 facts) and 197 entities (~40M characters, 400 facts). Due to computational costs, only one seed is used for these large-scale finetuning jobs on Gemini Pro.

**Evaluation.** To assess how well models learn and apply the finetuned information, we evaluate them across several task formats:

- **Direct QA:** Straightforward questions that explicitly ask about specific facts (e.g., "How many trails does Leif know about in Yosemite?"). While this format risks some train-test overlap due to limited question variation, it provides a baseline measure of fact retention.

- **Conversational QA:** Questions posed in a casual, friendly tone (e.g., "Hey, I was wondering - do you happen to know how many trails Leif is familiar with?"). This format tests generalization by avoiding literal overlap with training questions while maintaining similar semantic content.

- **Reasoning:** Problems that require applying finetuned knowledge within a broader reasoning chain, without explicitly asking for the fact (e.g., "If Leif wanted to hike half of his known trails this year, how many miles would that be?").

- **Wikipedia:** Two variants that test the model's ability to use finetuned knowledge in article-style text:
  - *Fill-in-the-blank:* Complete an article by inserting the correct fact into a designated blank
  - *Sentence completion:* Finish a truncated article using relevant finetuned information

- **Role Play:** Questions directed to the model as if it were the entity (e.g., "How many trails are you familiar with in the park?"), testing the model's ability to respond appropriately in-character.

These evaluation formats can be grouped into three broad categories:

- **QA-style:** Direct QA, Conversational QA, and Role Play - testing straightforward fact retrieval
- **Article-style:** Wikipedia fill-in and completion - testing integration with pretraining format
- **Reasoning-style:** Problems requiring fact application in broader contexts

To ensure evaluation quality, we generate three candidate questions for each fact and filter them using in-context learning (ICL) performance. Specifically, we provide an ICL model with the ground-truth information (not the training examples) and test if it can answer the questions correctly. We retain questions that the ICL model answers successfully (approximately 99% of cases) and randomly select one validated question per fact for the final evaluation set. While this validation process helps control for question difficulty and clarity, it has some limitations. For example, in reasoning tasks, the presence of relevant entity information in the ICL context may provide stronger hints about which facts to use compared to the finetuning scenario. However, this bias is consistent across experimental conditions and similar to the training setup.

