# OpenReview forum: "From Style to Facts: Mapping the Boundaries of Knowledge Injection with Finetuning"
_NeurIPS.cc/2025/Conference — NeurIPS 2025 poster_

### Official Review · Reviewer_KXho · 2025-06-16

**Clarity:** 3
**Significance:** 3
**Originality:** 2
**Rating:** 4
**Confidence:** 3

**Summary:**

This paper investigated the factors underlying the divergent fine-tuning performance between task customization and knowledge injection from a unified perspective. Specifically, it conducted large-scale fine-tuning experiments on Gemini v1.5 models, systematically manipulating data properties such as information quantity and type, entity type, and the format of both training and evaluation data. The experiments revealed nuanced results that help interpolate between the strengths and failure modes of fine-tuning.

**Questions:**

- The sentence between lines 309-313 is unclear in meaning. If there're no factual or logical errors, please consider rephrasing it for greater clarity and readability.
- The article/document format is an efficient way to store new knowledge. I wonder whether mixed fine-tuning could address the shortcomings of document/article-style data in knowledge injection. For example, one could combine articles containing new knowledge with QA-style examples derived from existing knowledge to balance knowledge injection and recall. I’d be curious to hear your perspective on this. Have you explored this mixed-format approach in your experiments, or are you aware of any existing work that investigates this direction?

**Ethical Concerns:**

["NO or VERY MINOR ethics concerns only"]

**Limitations:**

yes

**Paper Formatting Concerns:**

No major formatting issues.

**Quality:**

2

**Strengths And Weaknesses:**

### Strengths
- This paper carefully designed a testbed to evaluate two distinct task types: teaching tone for task customization and teaching Wikipedia facts for knowledge injection. The results revealed a stark contrast in performance, which supported the common belief that supervised fine-tuning primarily excels at "show, not tell" tasks but is less effective for incorporating new factual knowledge into the model.
- The paper next proposed several hypotheses to explain the large downstream performance gap and conducted experiments to test these hypotheses. The authors performed extensive fine-tuning experiments on Gemini v1.5 models, systematically ablating various data properties to evaluate their influence.
- The paper effectively controlled for randomness, significantly enhancing the reliability and validity of the results.
- The paper demonstrated the importance of the question-answer data format in fine-tuning for knowledge injection, a finding that aligns with prior work.
### Weaknesses
- The paper does not appear to include reasoning data for emotional information, which limits the scope of its evaluation. It would be helpful for the authors to explain why it was excluded.
- It's surprising that persona-related information is not easier for the model to learn compared to other entity types. This may be due to fine-tuning with small information volume. Did you experiment with fine-tuning on a greater number of persona-related facts to test whether data density plays a role?
- Some minor points:
	- The paper would benefit from a more detailed explanation of how the example interactions were rewritten to reflect different tone, such as whether the rewrites were done manually or automatically.
	- Including training loss curves for the LoRA fine-tuning experiments would be helpful. Did the loss converge to zero, or was there a clear plateau? This could shed light on how effectively the model fits the data.
	- The figure numbering appears to be incorrect in several places. Please review and correct the figure order.

---

> ### Author Response · Authors · 2025-07-31
>
> Thank you for your review! To address your questions:
> 1. The reason that we did not generate Reasoning tasks on emotional facts is that it’s not obvious how to design a multi-step logical reasoning problem using emotional facts. Given a numerical fact, we can generate an, e.g. arithmetic problem; given a categorical fact, we can generate an inverse problem; but emotional facts are too “simple” to create challenging reasoning questions, and so were omitted for the task. We will make sure to explicitly discuss this experiment design choice in the text.
> 2. Mixed fine-tuning indeed does help address the shortcomings of document/article-style data; this has been noted in existing work, e.g. Allen-Zhu, Li 2024, and also (but less directly) Ovadia et al. 2024 and Jiang et al. 2024. We see our results as providing a more concrete scientific understanding for why mixed fine-tuning has been observed to be so important.
> 3. “Did you experiment with fine-tuning on a greater number of persona-related facts to test whether data density plays a role?”
> Our experiments explicitly control for data density (when we compare between finetuning performance on different entity types, we ensure that all finetuning tasks involve the same information density). We agree this finding is counterintuitive, and took extensive measures to rule out potential confounders.
> 4. For the experiment in Section 3, rewrites in different tone were done automatically (we will add a note on this to make it explicit).
> 5. In all cases we ensured that the finetuning loss plateaus. The one exception where we saw a plateau significantly away from zero is when finetuning on a very large body of facts (the experiment in Figure 4.6); we believe this is indeed because the adapters lack the capacity to easily store that volume of information.

---

> ### Comment · Reviewer_KXho · 2025-08-06
>
> The authors basically answered my questions. I will maintain my original positive rating of the paper. Thanks.

---

### Official Review · Reviewer_u8cC · 2025-06-29

**Clarity:** 4
**Significance:** 4
**Originality:** 4
**Rating:** 5
**Confidence:** 4

**Summary:**

This paper conducts a large-scale empirical study of finetuning frontier language models (specifically Gemini v1.5) on datasets designed to interpolate between “task customization” (e.g., teaching tone or style) and “knowledge injection” (e.g., teaching new factual information). The authors systematically vary key factors, such as information type (numerical, categorical, emotional), training data format (question-answer pairs vs. Wikipedia-style articles), entity type (real-world, fictional, model personas), information quantity, and evaluation task alignment. Their experiments encompass over 4,000 finetuning settings, revealing clear trends about what works and what fails in knowledge injection. Most notably, they find that question-answer (QA) formatted training data consistently yields higher accuracy on Wikipedia-style factual tasks than Wikipedia-article formatted data, while the converse does not hold—a result that runs counter to much of the received wisdom and common practice in the community.

**Questions:**

1.	Theoretical Explanation of QA Superiority:
The finding that QA-format finetuning leads to higher accuracy on Wikipedia-style factual tasks (whereas the reverse is not true) is fascinating and goes against common expectations. Can the authors speculate, perhaps in the discussion, why this asymmetry exists? Are there connections to retrieval, prompt structure, or inductive biases in transformer models? Even a speculative or mechanistic discussion would strengthen the paper.
2.	Connection to Out-of-Context Reasoning:
The observed failure modes for knowledge injection, especially the challenge in applying injected knowledge to multi-step reasoning, appear closely related to recent work on out-of-context reasoning (e.g., Berglund et al., 2025). It would improve the discussion to explicitly connect your findings to this line of research, as both the empirical patterns and failure cases (e.g., QA format generalizing, but not Wikipedia) may have deeper roots in the model’s limitations around context and generalization.
3.	Future Work Directions:
Given the lack of theoretical explanation for several empirical patterns (e.g., QA superiority, numerical fact brittleness), what promising directions do the authors see for closing this gap? Could mechanistic interpretability or some training dynamics analysis be leveraged to better understand why QA data works so well? Are there experimental manipulations (e.g., varying the explicitness or structure of training data) that could tease apart the underlying mechanisms?
4.	Broader Impact on Model Editing/Updating:
The results have clear implications for the growing field of model editing and post-hoc updating of LLM knowledge. It would be useful to discuss how your findings could inform the design of knowledge editing pipelines, especially for dynamic or proprietary data injection.
5. "This is because the only nonsensical choice of training data for a finetuning model to learn about a persona it should play is where the examples demonstrate the model playing the persona." I find this sentence hard to read, can authors rephrase for clarity?
6. Both models evaluated in this work I'd assume are already instruction/RL finetuned, correct? So I wonder, is the results and conclusions specific to models that are already post trained, or do they hold true for pretrained base models?

Evaluation score could increase if the authors:

- Provide a deeper discussion of the mechanistic or theoretical reasons for their empirical findings, particularly the QA vs. article format asymmetry.

- More explicitly connect their results to the literature on out-of-context reasoning and situational awareness in LLMs.

- Investigate whether the conclusions still hold for pretrained base models that have not gone through post training.

Reference: Betley, J., Bao, X., Soto, M., Sztyber-Betley, A., Chua, J., & Evans, O. (2025). Tell me about yourself: LLMs are aware of their learned behaviors. arXiv preprint arXiv:2501.11120.

**Ethical Concerns:**

["NO or VERY MINOR ethics concerns only"]

**Final Justification:**

Thanks for authors' response. It's a very interesting paper, and the authors have address my comments, though some of them are more about future works that are probably beyond the scope of this paper. I'd keep my positive rating and vote for acceptance.

**Limitations:**

Yes, the authors are admirably candid about the limitations and open questions of their work. The absence of a theoretical account is acknowledged, and the societal/ethical aspects are not primary concerns for this methodological contribution.
I don't see much potential negative societal impact of this work.

**Quality:**

4

**Strengths And Weaknesses:**

Quality:
The paper’s experimental design is thorough, with careful controls for confounding factors such as information density, evaluation task difficulty, and entity type. The breadth of experiments and clarity in presenting the grid-search approach give the results strong empirical grounding. I especially appreciate the transparent reporting of negative results and failure modes, which help demystify common “rumors” and beliefs in the community around the brittleness and unpredictability of finetuning for knowledge injection.

Clarity:
The manuscript is well written and accessible, with clear figures and step-by-step explanations of experimental protocols. Key trends and takeaways are highlighted effectively. The clear distinction between types of knowledge, data formats, and task alignment makes the results actionable for practitioners.

Significance:
The paper addresses a critical and under-explored issue in LLM deployment: the practical limits and best practices for knowledge injection via finetuning. Its findings are highly relevant for anyone seeking to adapt LLMs to post-training facts, proprietary data, or changing world knowledge. The empirical demonstration that QA data outperforms article data in knowledge injection, even for article-like test tasks, is a valuable contribution with direct practical impact.

Originality:
While previous works have noted the brittleness of finetuning for knowledge injection, this paper stands out for its systematic comparison of data formats and task alignments at scale, and for dispelling the myth that finetuning is simply harder for factual knowledge than for style. The nuanced findings around information type (numerical vs. categorical/emotional) and the failure modes for multi-step reasoning further deepen our understanding.

Weaknesses:

The most intriguing empirical finding—that QA-format finetuning generalizes to Wikipedia tasks much better than the reverse—is not given a strong theoretical explanation. The authors acknowledge this but do not provide a mechanistic hypothesis, leaving an important gap for follow-up work.

The connection to broader issues of out-of-context reasoning and generalization is hinted at but not fully developed (see below for suggestions).

---

> ### Author Response · Authors · 2025-07-31
>
> Thank you for your review! To address your questions:
> 1. We agree this is a fascinating phenomenon and would make sure to include an extended discussion in the additional page that camera-ready provides. Our hypothesis is that this is closely related to other observations made by the community (e.g. in e.g. Allen-Zhu and Li [2024]) that the presence of QA in pretraining data is critical for developing general recall abilities. It’s not clear to us what a mechanistic explanation might be (to the best of our knowledge, we are not aware of any concrete theories on this by the community either), but one possibility is indeed that this is an artifact of some inductive bias of transformers (e.g., towards induction heads that most readily parse QA-style sequences).
> 2. We agree in the paper that there does appear to be a strong connection with Berglund et al. 2023, 2024, but will extend our discussion on this (which is currently scattered between L90, L98, L260, and L324).
> 3. We think mechanistic interpretability is probably the most promising route to developing useful theories for these patterns. Since theory does not appear to provide an obvious answer, our hypothesis is that these phenomena (e.g. QA superiority) are attributable to mechanistic causes. We expect that testing different interventions when training models on synthetic data (e.g., as is done in the Phi-series) may yield some useful clues here.
> 4. While we currently cite knowledge editing literature for their insights on parametric model knowledge, we agree that an extended discussion on the implications of our results for model editing would be useful and will revise accordingly.
> 5. We will correct this sentence: "This is because the only nonsensical choice of training data for a finetuning model to learn about a persona it should play is where the examples demonstrate the model playing the persona." -> "When fine-tuning a model to adopt a persona, the only sensible training data would be examples in which the model is already acting out that persona."
> 6. Both models in this work have already undergone post-training. We would anticipate that similar trends hold, but with the caveat that models which have not undergone instruction tuning would likely perform exceptionally poorly at answering some of the less conventional tasks we evaluate on, e.g. the reasoning tasks. We will include a more detailed discussion on some of the caveats here, but agree this is a promising avenue for future work.

---

> > ### Comment · Reviewer_u8cC · 2025-08-06
> >
> > Thanks for authors' response. It's a very interesting paper, and the authors have address my comments, though some of them are more about future works that are probably beyond the scope of this paper. I'd keep my positive rating and vote for acceptance.

---

### Official Review · Reviewer_F74L · 2025-07-02

**Clarity:** 2
**Significance:** 3
**Originality:** 3
**Rating:** 4
**Confidence:** 3

**Summary:**

This paper offers a comprehensive examination of why finetuning sometimes excels at stylistic “task customization” yet often falters at factual “knowledge injection.” The authors argue that the two goals are conceptually the same and probe this claim through more than 4 000 finetuning runs on Gemini v1.5 Pro and Flash. Their grid varies five axes—entity class (real, fictional, persona), information quantity (20/200/4 000 facts), information type (numeric, categorical, emotional), training‑data format (QA, multi‑turn QA, chain‑of‑thought, role‑play, Wikipedia‑style articles) and evaluation task—while carefully controlling for data size, fact difficulty, and safety. A mirrored corpus of real and synthetic entities eliminates distribution‑shift confounds.

The study reveals several consistent patterns. Alignment between training format and evaluation task is the strongest predictor of success; plain QA data generalize well, whereas Wikipedia‑like documents are least effective. Numeric facts are harder to retain than categorical or affective ones, and accuracy decays roughly as a power law with the number of facts injected. Crucially, finetuning personas is not significantly easier than finetuning knowledge about real or fictional entities, dispelling the idea that self‑referential styles are inherently simpler.

By disentangling these variables, the authors demystify the boundaries of parametric knowledge storage. Their findings provide actionable guidance—favor QA‑style datasets, beware reasoning tasks’ “random‑access” and “reversal‑curse” limits, and complement large fact sets with retrieval or prompt engineering. The work establishes a rigorous benchmark and reframes the community’s intuition about what finetuning can—and cannot—accomplish.

**Questions:**

**1.  Please clarify the distinction between “Wiki” and “Wiki Fill in” in the figures**

Suggestion: Explicitly define in the legend or main text how these two formats differ in fill in approach, or evaluation protocol.
Scoring Criteria: If the authors provide clear definitions and examples, my Clarity score can rise from 2 to 3.

**2.  Please explain the evaluation differences between Gemini v1.5 Pro and Flash**

Question: Why does the Pro variant cover only QA, multi turn QA, and role play (RP), while Flash covers QA, BP, Wiki, and Reasoning?
Suggestion: List the exact task sets and naming conventions for each variant, and justify the choice of tasks and their impact on results.
Scoring Criteria: If the authors present a unified taxonomy or comparative analysis, my Quality or Clarity scores can increase to 4; otherwise, those dimensions may decrease.

**3.  Why was Emotional information not evaluated on Reasoning tasks?**

Suggestion: Either add corresponding reasoning experiments for emotional info or discuss in the limitations section why it was omitted and how this affects conclusions.
Scoring Criteria: If emotional reasoning results are added or reasons are explained, my Quality score can go from 3 to 4.

**4.  Please provide error bar calculation and statistical testing details**

Question: How are the error bars computed? Is a normality assumption used? Why are there no confidence intervals or significance tests?
Suggestion: Include the error bar formula, results of normality checks, and at least one form of statistical significance analysis (e.g., p values or confidence intervals).
Scoring Criteria: If full statistical details are provided, my Quality score can rise to 4; otherwise, Quality may decline further.

**Ethical Concerns:**

["NO or VERY MINOR ethics concerns only"]

**Limitations:**

**Risk of misinformation and misuse**

While the appendix provides examples of the model confidently “hallucinating” facts post‑injection, the main text does not analyze these error modes nor propose any mitigation or detection strategies.

**Suggestions**

Add a dedicated “Limitations” section that systematically covers the above points.

**Paper Formatting Concerns:**

No concerns

**Quality:**

3

**Strengths And Weaknesses:**

**Strengths**
1. Scale and Rigor
Runs 4 000+ finetuning jobs on Gemini v1.5 (Pro/Flash) across a multi-grid—much more exhaustive than prior single factor studies.
Uses a paired real / fictional entity set with matched fact density and safety filtering, effectively removing confounds.
Places knowledge injection and task customization in one experimental frame and empirically shows they are not inherently different in difficulty.
2. Dataset & Benchmark Contribution
Releases multi format datasets and LoRA configs, offering a reusable, industry grade benchmark.

**Weaknesses**
1. Statistical Under specification
Error bars are shown but not explained (apparently ±1 σ); no normality checks, confidence intervals, or significance tests.
2. Sparse Interpretation
Several acronyms (e.g., Wiki, Wiki-Fill-in) are undefined at first mention, increasing cognitive load.
Heat maps get brief commentary, with little discussion of anomalies or negative cases.
3. Limitations
Evaluation is confined to the Gemini v1.5 family; cross model generalization is untested.
Benchmarks focus on closed form QA, omitting open ended generation and long form writing tasks.

---

> ### Author Response · Authors · 2025-07-31
>
> Thank you for your review! To address your questions:
>
> 1. The difference between the Wiki and Wiki-Fill-In tasks is described on L193-L196: in “Wiki-Fill-In” (Wikipedia fill-in-the-blank) tasks models are asked to fulfill a masked prediction task (a few words in the middle of the article are covered up and must be predicted), whereas in “Wiki” (Wikipedia sentence completion) tasks models are asked to fulfill an autoregressive task (the last sentence is ended early and its ending must be predicted). We will add pointers to L193-L196 in more places where these tasks are mentioned, and will also add examples.
> 2. We ran all experiments on both Pro and Flash variants. We believe the confusion may have arisen because we only plotted the performance of Pro and Flash on disjoint sets in the main text of the paper (due to page constraints). The full experiment results can always be found in the Appendix (starting at L430). We will make sure to add pointers to address this.
> 3. The reason that we did not generate Reasoning tasks on emotional facts is that it’s not obvious how to design a multi-step logical reasoning problem using emotional facts. Given a numerical fact, we can generate an, e.g. arithmetic problem; given a categorical fact, we can generate an inverse problem; but emotional facts are too “simple” to create challenging reasoning questions, and so were omitted for the task. We will make sure to explicitly discuss this experiment design choice in the text.
> 4. Error bars indicate standard error. Our error bars indicate accuracy, which is a binomial random variable and normality is immediate (CLT). We will make sure to mention that bars indicate standard error.

---

> > ### Comment · Reviewer_F74L · 2025-08-05
> >
> > Thanks for your response. I would keep my positive rating.

---

### Official Review · Reviewer_sHAo · 2025-07-05

**Clarity:** 3
**Significance:** 3
**Originality:** 3
**Rating:** 4
**Confidence:** 4

**Summary:**

The paper conducts large-scale empirical studies to investigate how supervised fine-tuning with LLMs learns task customization easily but barely injects any knowledge. The authors design a spectrum of knowledge injection settings by identifying five different axes: information quantity, information type, entity type, training data format, and evaluation task. Experiment results with Gemini-v1.5 show that question-answering training data is the most effective format for knowledge injection, categorical information is relatively easier to learn, and it is hard for LLMs to leverage fine-tuned knowledge during multi-step reasoning.

**Questions:**

Q1: In the abstract (Line 6), what do you mean by "a distinction without a difference"?
Q2: There is a broken link in Line 79.
Q3: Can you switch to using \citep when making citations in the paper?
Q4: Can you break down Section 4.1 into several subsections and highlight your major findings?
Q5: Please reorder the sentences in Lines 148-153.

**Ethical Concerns:**

["NO or VERY MINOR ethics concerns only"]

**Final Justification:**

The issues raised in my review have been largely resolved. Therefore, I vote for the acceptance of this paper.

**Limitations:**

Yes.

**Quality:**

3

**Strengths And Weaknesses:**

S1: The research question studied in this paper is interesting and significant. Although superficial alignment has been reported in many papers, there lack studies thoroughly investigating the underlying factors.
S2: The experiments in the paper are comprehensive and thorough, which require a great amount of efforts.
S2: The writing in the paper is mostly clear and easy to follow.

---
W1: In Section 3, how did the authors extract facts from Wikipedia articles? Can you include a manual verification to avoid any hallucinations during this process?
W2: In Section 4, how did the authors set the hyperparameters for fine-tuning Gemini-v1.5? Which model did you use to generate or augment the fine-tuning data?
W3: Can you repeat the experiments for another model family? Currently, it is unclear if the findings generalize to model series other than Gemini-v1.5. In addition, can you report the zero-shot and in-context learning performance for those evaluation tasks without any fine-tuning?

---

> ### Author Response · Authors · 2025-07-31
>
> Thank you for your review! To address your questions:
> 1. We used a language model to extract facts from the Wikipedia articles and then performed manual verification. Regarding hallucinations, we did not identify any instances—we have also open sourced our entire pipeline and artifacts to allow for outside scrutiny.
> 2. For learning rate and number of adapters, the hyperparameters we chose are exactly the default values recommended in the Gemini v1.5 finetuning guidance. We chose the number of iterations (100) to be “conservatively large”, i.e. large enough that we do not encounter undertraining failures.
> 3. Our choice to run experiments on Gemini v1.5 Flash and Gemini v1.5 Pro are due to resource constraints and the significant cost of the experiments. The zero-shot and in-context learning rates for *all tasks* in Section 4 are <1% and >90% respectively; we will add these figures into the paper for context and appreciate the suggestion!
> 4. "A distinction without a difference" is an idiom / name of a fallacy where one draws a difference between two things that aren’t clearly different.
> 5. Q2-Q5: Thank you for the suggestions! We have incorporated them.

---

> > ### Comment · Reviewer_sHAo · 2025-08-04
> >
> > Thanks for your response. I would keep the rating and vote for acceptance of the paper.

---

### Decision · Program_Chairs · 2025-09-17

**Decision:**

Accept (poster)

**Comment:**

**Summary**
This paper investigates the differential effectiveness of supervised fine-tuning for task customization versus knowledge injection in LLMs. Through over 4,000 fine-tuning experiments on Gemini models, the authors systematically vary five axes: information quantity, information type (numeric, categorical, emotional), entity type (real, fictional, persona), training data format (QA, multi-turn QA, chain-of-thought, role-play, Wikipedia-style), and evaluation task.

Experiments reveal four interesting takeaways. First, QA formatted data is most effective for knowledge injection, even outperforming Wikipedia-style articles on article-completion tasks. Second, categorical and emotional information is easier to inject than numeric facts. Third, fine-tuned knowledge does not reliably transfer to multi-step reasoning tasks. Finally, persona customization is not inherently easier than factual knowledge injection when controlling for information density.

**Strengths**
Offers interesting insight into what type of data and format work well for LLM finetuning. Experiments systematically controlled for potential confounds like information density and fact difficulty across a multi-dimensional grid of variables.

**Weaknesses**
Lack of theoretical explanation for: 1) empirical findings, e.g., merit of QA-formatted data; 2) empirical performance of LLMs beyond Gemini-v1.5 model family tree; and 3) thoroughness/clarity in statistical reports.

**Reason for Accept/Reject**
Substantial empirical contribution, and convincing author response to reviewer concerns.

**Rebuttal Discussion**
Concern was raised about how facts were extracted from Wikipedia and the potential for hallucinations. The authors clarified that LLM was used for extraction, followed by manual verification.

Reviewers also requested clarifications on the distinction between evaluation tasks (e.g., Wiki vs. Wiki-Fill-In), the coverage of different models (Pro vs. Flash), and the omission of reasoning tasks for emotional facts. The authors provided detailed explanations with grounded references to paper writing, and justified the design choice for emotional facts to difficulty in formulating multi-step problems.

Theoretical explanation for the QA-format superiority remains unresolved, but the authors proposed a speculative connection to known inductive biases in transformers. They also agreed to more explicitly connect their work to related research on out-of-context reasoning and model editing.